# Cold Atmospheric Plasma Restores Paclitaxel Sensitivity to Paclitaxel-Resistant Breast Cancer Cells by Reversing Expression of Resistance-Related Genes

**DOI:** 10.3390/cancers11122011

**Published:** 2019-12-13

**Authors:** Sungbin Park, Heejoo Kim, Hwee Won Ji, Hyeon Woo Kim, Sung Hwan Yun, Eun Ha Choi, Sun Jung Kim

**Affiliations:** 1Department of Life Science, Dongguk University-Seoul, Goyang 10326, Korea; do31100@dongguk.edu (S.P.); heejoo0923@dongguk.edu (H.K.); hweewon96@dongguk.edu (H.W.J.); opopr5@dongguk.edu (H.W.K.); skskbby@dongguk.edu (S.H.Y.); 2Plasma Bioscience Research Center, Kwangwoon University, Seoul 01897, Korea; ehchoi@kw.ac.kr

**Keywords:** apoptosis, cold atmospheric plasma, breast cancer, genome-wide expression, reactive oxygen species

## Abstract

Paclitaxel (Tx) is a widely used therapeutic chemical for breast cancer treatment; however, cancer recurrence remains an obstacle for improved prognosis of cancer patients. In this study, cold atmospheric plasma (CAP) was tested for its potential to overcome the drug resistance. After developing Tx-resistant MCF-7 (MCF-7/TxR) breast cancer cells, CAP was applied to the cells, and its effect on the recovery of drug sensitivity was assessed in both cellular and molecular aspects. Sensitivity to Tx in the MCF-7/TxR cells was restored up to 73% by CAP. A comparison of genome-wide expression profiles between the TxR cells and the CAP-treated cells identified 49 genes that commonly appeared with significant changes. Notably, 20 genes, such as KIF13B, GOLM1, and TLE4, showed opposite expression profiles. The protein expression levels of selected genes, DAGLA and CEACAM1, were recovered to those of their parental cells by CAP. Taken together, CAP inhibited the growth of MCF-7/TxR cancer cells and recovered Tx sensitivity by resetting the expression of multiple drug resistance–related genes. These findings may contribute to extending the application of CAP to the treatment of TxR cancer.

## 1. Introduction

Recurrence of cancer due to the acquisition of resistance to chemotherapy remains a serious obstacle for the clinical treatment of cancer patients [1]. A high ratio of cancer patients who receive chemotherapy acquire drug resistance [2,3]. Among breast cancer patients, an estimated one in three will eventually develop recurrent or metastatic disease, causing poor prognosis with a median 5-year survival of <25% [4]. Paclitaxel (Tx) has been widely used to treat a variety of cancer types, including breast cancer [5]. Tx binds to β-tubulin subunits of the microtubule, preventing them from undergoing the depolymerization process, which is crucial during the course of mitosis [6]. Subsequently, the dissociation of the spindle is inhibited, and therefore, the cell cycle is blocked in the G2/M phase and apoptosis [7].

A few molecular mechanisms have explained Tx resistance (TxR). Altered microtubule physiology is a major dominator escaping the cytotoxicity of the drug [8]. Moreover, the HER2 signaling cascade influences several routes of resistance, such as drug efflux and drug metabolism [9]. HER2 overexpression was found not only in cultured TxR cells [10] but also in TxR cancer patients [11]. A cascade of protein kinases involving ERK1/2 is activated by HER2 via phosphorylation, and then target genes such as cyclin A, cyclin B, and CD44 are induced to enhance cell proliferation and stemness [12]. TxR can also be conferred by the upregulation of drug efflux pumps, such as ABCB1 [13] and MRP3 [14], which are often overexpressed in HER2-overexpressing tumors. Strategies to circumvent TxR include blockades of ABCB1 and MRP efflux and the development of systemic agents with low susceptibility to common resistance mechanisms, such as HER2 overexpression [15]. Despite these works to eradicate TxR cancer, a high percentage of TxR cancers still remain aggressive and threaten the prognosis of cancer patients [16].

Cold atmospheric plasma (CAP) has received attention from basic biological science, as well as medical researchers, due to its ability to specifically induce the death of cancer cells over normal cells [17,18]. A wide spectrum of cancer cells, including breast, ovary, liver, lung, brain, and skin cancer cells, underwent apoptosis, suggesting the potential of CAP as an alternative cancer treatment option [19,20]. Given the heterogeneity of cancer cells, including drug-resistant cancer cells, it is hoped that CAP is also able to induce apoptosis of these cells and even to help recover sensitivity to the therapeutic chemicals. In previous studies, CAP returned the temozolomide-resistant glioblastoma cells [21] and cisplatin-resistant ovarian cancer cells [22] to a drug-sensitive state, in addition to inducing apoptosis. However, little is known about the molecular mechanism by which CAP resets the protein expression levels of resistant cancer cells to those of their parental cancer cells. A study identified the “cell death/survival and cancer” network as the pivotal pathway that was deregulated by CAP for the tamoxifen-resistant MCF-7 breast cancer cells [23]. In particular, MX1 and HOXC6 genes were identified to be directly involved during the acquisition of tamoxifen resistance and recovery of sensitivity by CAP. With this limited information, however, it is premature to conclude that CAP recovers sensitivity to diverse therapeutics in drug-resistant cancers.

In this study, we explored the potential of CAP to inhibit the proliferation of cancer cells carrying a more extended range of drug resistance using MCF-7/TxR. Next, the gene signature responsible for the resistance and sensitivity to Tx was identified at the genome-wide level. The expression of selected genes that were deregulated by Tx and CAP was further examined at the protein level.

## 2. Results

### 2.1. CAP Recovered Tx Sensitivity in MCF-7/TxR Cells

MCF-7/TxR cells were developed by continually exposing the MCF-7 cells to Tx of step-wise increased concentrations up to 60 ng/ml in the culture media. The MCF-7/TxR cells produced slightly higher levels of reactive oxygen species (ROS) compared to the parental MCF-7 cells, while both cell types responded to CAP with increased ROS levels (Figure 1). To determine whether the recovery of sensitivity by CAP comes from any decreased pumping-out activity of the cell for chemicals, fluorescence-tagged Tx and doxorubicin, which itself emits fluorescence, were used, and then the fluorescence was measured by Fluorescence Activated Cell Sorter (FACS). As a result, no significant difference was found for either chemical in the CAP-treated MCF-7/TxR cells compared to the non-treated control cells, indicating that the sensitivity recovery is not from any change in the drug transport rate across the plasma membrane (Figure 2).

The potential of CAP to recover the MCF-7/TxR cells’ sensitivity to Tx was monitored by two experimental approaches. First, the cells were treated with CAP followed by Tx in amounts of 30 and 60 ng/mL. Then, the survival of cells was examined by a colony formation assay (Figure 3A and Appendix A). MCF-7/TxR cells proliferated more quickly than MCF-7, but the proliferation was suppressed by CAP. Notably, CAP treatment reset the resistant cells’ sensitivity to Tx in a dose-dependent manner. When the CAP-treated MCF-7/TxR cells were treated with Tx of 60 ng/mL, their growth decreased by 73%, while that of the non-treated cells decreased by only 50%. Second, the effect of CAP on sensitivity recovery was examined by tracking the growth of the cells for 5 days using a dye-based assay. The result also indicated a higher growth rate for the MCF-7/TxR cells (Figure 3B) and recovery of drug sensitivity when the cells were treated with CAP (Figure 3C). All these data support the fact that CAP sets the state of drug resistance back to the sensitive state, enabling Tx to induce the death of the chemo-resistant cancer cells.

### 2.2. Expression of a Set of Genes Is Reversed from MCF-7 via MCF-7/TxR to CAP-Treated MCF-7/TxR Cells

To investigate the molecular mechanism of CAP for the sensitivity recovery, a genome-wide expression array analysis was performed. The array covering 58,201 human genes was analyzed in duplicate for each set of MCF-7 vs. MCF-7/TxR and MCF-7/TxR vs. CAP-treated MCF-7/TxR. With the cut ratio higher than 1.3 fold, 1335 genes showed expression differences in the MCF-7 vs. MCF-7/TxR and 367 genes in the MCF-7/TxR and MCF-7/TxR vs. CAP-treated MCF-7/TxR, representing 49 genes that appeared in both sets (Figure 4A). Finally, 20 genes showed the opposite alteration during the course from MCF-7 via MCF-7/TxR to CAP-treated MCF-7/TxR (Appendix A). The expression of genes from the array data was re-examined by qPCR for six genes that were selected from the 20 genes in Figure 4A, and the result confirmed the same alteration by Tx and CAP (Figure 4B).

With the 1335 genes from the MCF-7 vs. MCF-7/TxR, the Ingenuity Pathway Analysis (IPA) network analysis was performed, and this represented “Nutritional Disease, Organismal Injury and Abnormalities, Carbohydrate Metabolism” as the top network (Figure 5A). Notably, TGF-β1 comprises a hub of the network through interacting with many genes regulated by TGF-β1, such as TLE4, PLEK2, and CPQ. Meanwhile, the network of the 367 genes from MCF-7/TxR vs. CAP-treated MCF-7/TxR represented “Embryonic Development, Nervous System Development and Function, Organ Development” as the top network (Figure 5B). In the network, the ERK1/2 hub was notable, showing interaction with multiple genes, including DLK1, SPRY4, and APH1A.

### 2.3. CEACAM1 and DAGLA Are Regulated during the Restoration of Sensitivity to Tx by CAP

Even though 20 genes were deregulated during the acquisition of drug resistance and recovery, their contribution to these processes had to be determined. To address this, we selected and focused on two genes: CEACAM1 from the downregulated group and DAGLA from the upregulated group in the CAP-treated MCF-7/TxR cells. After having confirmed the deregulation of gene expression found in the microarray for the two genes by qPCR, their expression was further examined by Western blot analysis. As a result, the profile of protein expression was similar to that of RNA expression (Figure 6A,B). In detail, expression of DAGLA was decreased in the MCF-7/TxR compared to MCF-7 but increased by CAP. In the case of CEACAM1, the expression profile was the opposite (i.e., increased in the MCF-7/TxR but decreased by CAP). Notably, the effect of CAP on the protein expression of the two genes was deteriorated by NAC, an ROS inhibitor (Figure 6C,D).

Next, the effect of the two genes on drug sensitivity was monitored at their dysregulated conditions through colony formation assay. When DAGLA was downregulated by siRNA, MCF-7/TxR proliferated at a higher rate up to 40% than the control siRNA-treated cells (Figure 7A and Appendix A). Meanwhile, lower cell growth up to 8% compared to control was observed when CEACAM1 was downregulated (Figure 7B and Appendix A). However, all the siRNA-treated cells did not show significant change of sensitivity to Tx, even though a few samples such as 30 ng/mL of Tx- and siRNA-treated cells for DAGLA and CEACAM1 showed difference up to 1.5 fold compared to control.

## 3. Discussion

This study aimed to evaluate the potential of CAP to overcome TxR and thereby prevent cancer recurrence in a cell model system. The underlying molecular mechanism of sensitivity recovery was investigated through tracking the change of genome-wide expression during the acquisition of TxR and during the recovery of sensitivity by CAP. Notably, the level of ROS in the MCF-7/TxR cells was higher than in MCF-7, and the difference became more significant when the cells were exposed to CAP. This is reminiscent of the fact that drug-resistant cancer cells in general produce higher levels of ROS than their parental cancer cells [24]. An increase of ROS in cells can contribute to a higher growth rate, but levels higher than a threshold can induce cell death or apoptosis [25]. Therefore, the increased ROS levels in the drug-resistant cells may serve as a double-edged sword: they may increase the cells’ proliferation rate but make them more vulnerable to CAP. In fact, the MCF-7/TxR cells showed a higher growth rate than MCF-7 and a higher level of death induction by CAP.

In this study, CAP did not affect the net intracellular level of Tx (i.e., transport of Tx across the plasma membrane after CAP treatment was not changed significantly). In contrast, the expression of genes previously known to be responsible for drug resistance, such as Bcl2L13 (1.86-fold decrease) [26,27] and HOXA9 (2.14-fold increase) [28], was changed by CAP. This discrepancy explains the complication of drug transport in the MCF-7/TxR cells. Thus, genes in pathways associated with processes other than transport may have been affected during the acquisition of TxR. In fact, among the gene signatures identified in MCF-7 vs. MCF-7/TxR and MCF-7/TxR vs. CAP-treated MCF-7/TxR, only a portion shared common pathways and genes. Elucidation of the molecular mechanism of the differentially expressed genes could contribute to constructing the scenario of how TxR emerges and to establishing how to manipulate the TxR cancer. Notably, TGF-β took part in the central hub of the IPA network constructed with deregulated genes in the MCF-7/TxR cells. In addition, canonical pathway analysis identified the “interferon signaling”, “Wnt/β-catenin”, and “iNOS signaling” pathways as the most significant pathways. All these are pivotal to drive cellular differentiation and cancer development [29,30,31].

CEACAM1 is a glycoprotein belonging to the carcinoembryonic antigen (CEA) family that is expressed in a wide variety of cells [32]. The prognostic value of CEACAM1 in cancer is controversial: CEACAM1 expression correlates with good prognosis in mammary carcinomas, whereas in melanomas, upregulation of CEACAM1 is accompanied by poor overall survival [33,34]. It is notable that CEACAM1 was upregulated in the MCF-7/TxR cells and downregulated by CAP. Meanwhile, DAGLA was downregulated in the MCF-7/TxR cells but upregulated by CAP. DAGLA catalyzes the hydrolysis of diacylglycerol to 2-arachidonoylglycerol and free fatty acids [35]. So far, little has been found regarding its possible role in human tumorigenesis. A study indicated the upregulation of DAGLA in oral cancer [36]. However, no information is available on other cancers, including breast cancer. Database analysis identified the downregulation of DAGLA in breast cancer tissue compared to normal tissues (Appendix A). In addition, breast cancer patients with higher expression of DAGLA showed better prognosis than those with lower expression, suggesting DAGLA as a tumor suppressor in breast cancer. In the case of CEACAM1, no significance was found in either expression or prognosis between normal and cancer tissues (Appendix A). CAP has previously shown to recover sensitivity to tamoxifen, however, remarkably different gene and network signature for Tx resistance was identified by genome-wide expression analysis, suggesting a differential molecular mechanism of CAP acting on the Tx-resistant cancer cells from the tamoxifen-resistant cancer. It should be mentioned that DAGLA and CEACAM1 were deregulated while the sensitivity was recovered by CAP, however, they are not directly responsible for the sensitivity recovery because downregulation with siRNA did not induce significant change of sensitivity to Tx. Validating other genes from the microarray data (Figure 4A) is needed to identify genes which are pivotal to the recovery of sensitivity. Altogether, these data indicate that CAP recovers Tx sensitivity at least in part by modulating the expression of oncogenes or tumor suppressors.

A limitation of this study is the lack of verification of the in vitro data on in vivo platforms, such as an animal model. Applying CAP treatment in combination with Tx to tumor tissues formed from xenografted MCF-7/TxR cells and tracking the molecular change could clearly demonstrate the potential of CAP to recover Tx sensitivity. Also, using only the ROS inhibitor obscures defining the specific casts in CAP that regulate the identified genes. Further study should include finely tuning the CAP treatment conditions, including power strength and duration, to overcome obstacles while applying it to in vivo systems.

## 4. Materials and Methods

### 4.1. Cell Culture and CAP Treatment

The MCF-7 human breast cancer cell line was purchased from the American Type Culture Collection (ATCC) (Manassas, VA, USA) and cultured in RPMI-1640 medium (Gibco, Los Angeles, CA, USA) supplemented with 10% fetal bovine serum (Capricorn, Ebsdorfergrund, Germany) and 2% penicillin/streptomycin (Capricorn) at 37 °C in a humidified incubator containing 5% CO_2_. The MCF-7/TxR cell line was generated by sequential exposure to increasing doses of Tx (Sigma-Aldrich, St. Louis, MO, USA) up to 60 ng/mL The mesh-dielectric barrier discharge (DBD) type of CAP device was produced at the Plasma Bioscience Research Center (Kwangwoon University, Seoul, Korea). The surface of culture media was treated with CAP from a 4 mm distance 10 times each for 30 s 1 h apart (10 × 30 s) with the energy of 0.3 kV and 12.9 kHz at an argon gas flow of 1.8 L/min.

### 4.2. Colony Formation and Cell Proliferation Analysis

For the colony formation assay, 3 × 10^3^ cells of MCF-7 and MCF-7/TxR were seeded in 60 mm culture dishes with 2 mL medium and treated with CAP followed by exposure to either Tx (Sigma-Aldrich) or DMSO as a vehicle control 24 h later. siRNAs were transfected 24 h before CAP treatment. After incubating the cells for 14–21 days, colonies were fixed with methanol/acetic acid (7:1) and stained with 0.2% crystal violet solution (Sigma-Aldrich). The colony area was analyzed using the ImageJ software program [37]. For the cell proliferation assay, 3 × 10^3^ cells were seeded per well of a 96-well plate with 100 μL medium and 24 h later treated with CAP and Tx. The cell growth rate was monitored at specific time points using CCK-8 solution (Enzo, New York, NY, USA) following the supplier’s protocol.

### 4.3. Reactive Oxygen Species (ROS) Detection

For ROS detection, 5 × 10^5^ cells seeded in a well of a 6-well plate were treated with CAP by the 10 × 30 s scheme. The cells were treated with 20 μM of DCFH-DA (Sigma-Aldrich) for 30 min in a humidified incubator at 37 °C 24 h later. The ROS level was calculated by measuring fluorescence using an Infinite 200 Pro fluorescence plate reader (Tecan, Mannedorf, Switzerland). To inhibit ROS synthesis, N-acetylcysteine (NAC) (Sigma-Aldrich) was added in a final concentration of 2 mM 2 h before CAP treatment.

### 4.4. Microarray Analysis

For the microarray analysis, 3 × 10^5^ cells of MCF-7, MCF-7/TxR, and CAP-treated MCF-7/TxR were seeded in 60 mm culture dishes and cultured for 24 h before harvest. Total RNA was extracted using a ZR-Duet DNA/RNA MiniPrep kit (Zymo Research, Irvine, CA, USA). Labeled cRNA was synthesized from 600 ng of RNA and hybridized on SurePrint G3 Custom Gene Expression Microarrays, 8 × 60K (Agilent, Santa Clara, CA, USA) following the Agilent One-Color Microarray-Based Gene Expression Analysis protocol (Agilent, V 6.5, 2010). The hybridized array was analyzed using an Agilent SureScan Microarray Scanner. All array data were uploaded to the Gene Expression Omnibus (GEO) database, and they can be accessed via their website (http://www.ncbi.nlm.nih.gov/geo/; Series accession number GSE131480).

### 4.5. Pathway and Clustering Analysis

Pathway analysis was performed using the Ingenuity Pathway Analysis (IPA) software (Qiagen, Redwood City, CA, USA) by submitting gene pools comprised of genes showing expression change ≥ 1.3 and *p*-value < 0.05. Clustering analysis was performed using the Clustering 3.0 software (http://bonsai.hgc.jp/~mdehoon/software/cluster/) and then visualized using the TreeView program (http://jtreeview.sourceforge.net/).

### 4.6. Quantitative RT-PCR (qPCR)

Total RNA was extracted from cells 24 h after CAP treatment using the ZR-Duet DNA/RNA MiniPrep kit (Zymo Research) and reverse transcribed to cDNA using ReverTra Ace qPCR RT Master Mix with gDNA Remover (Toyobo, Osaka, Japan). DNA was amplified using KAPA SYBR FAST qPCR Kit Master Mix ABI Prism (Kapa Biosystems, Wilmington, MA, USA) on an ABI 7300 instrument (Applied Biosystems, Foster City, CA, USA). The relative gene expression level was calculated using the 2^−ΔΔCt^ method with glyceraldehyde-3-phosphate dehydrogenase (GAPDH) as an internal control. The PCR condition was as follows: denaturation at 95 °C for 3 min, 40 cycles of denaturation at 95 °C for 3 s, and annealing/extension at 60 °C for 40 s. Primer sequences used for qPCR are listed in Appendix A.

### 4.7. Western Blot Analysis

To extract total proteins from cultured cells, the cells were collected and suspended in ice-cold RIPA lysis buffer with a protease inhibitor cocktail (Thermo Fisher Scientific, Waltham, MA, USA). After determining protein concentration by BCA assay (Thermo Fisher Scientific), 50 μg of the protein was subjected to SDS-PAGE. After electrophoresis, the proteins were transferred onto PVDF membranes (Whatman, Maidstone, UK), and then the membranes were soaked in blocking solution (5% non-fat milk diluted in 0.1% Tween-20 TBS) for 1 h at room temperature. The blots were incubated overnight at 4 °C with anti-CEACAM1 (1:500, Bioss, Woburn, MA, USA) and anti-DAGLA (1:500, Bioss) antibodies. The blots were additionally probed with an anti-β-actin antibody (1:800, Bioss) as an internal reference. After incubation with HRP-conjugated goat/rabbit anti-rabbit secondary antibodies (1:1000, GeneTex, Irvine, CA, USA) for 1 h, protein bands were visualized with ECL reagent (AbFrontier, Seoul, Korea) and quantified with Image Lab software (Bio-Rad, Hercules, CA, USA). The whole blot figures can be accessed in Appendix A).

### 4.8. Fluorescene Activated Cell Sorter (FACS) Analysis

Apoptosis and drug transport were evaluated by FACS. For the apoptosis analysis, 1 × 10^6^ cells were seeded in a 60 mm dish, treated with CAP, and cultured for 24 h. 1 × 10^5^ washed cells were treated with 5 μL of FITC Annexin V and 5 μL of propidium iodide (PI) using an FITC Annexin Apoptosis Detection Kit (BD Technologies, Franklin Lakes, NJ, USA). Samples were analyzed using a FACS Canto II flow cytometer (BD Technologies). To monitor drug uptake by cells, 1 × 10^6^ cells seeded in a 60 mm dish were treated with Flutax (Santa Cruz Biotechnology, Santa Cruz, CA, USA, sc-203958) for 4 h or doxorubicin (Cayman Chemical, Ann Arbor, MI, USA, 15007) for 24 h, respectively, at final concentration of 10 μM. Fluorescene was detected with FACSAria III (BD Technologies) and analyzed with BD FACSDiva software.

### 4.9. Statistical Analysis

Student’s t-test was applied to compare gene expression levels between CAP-exposed MCF-7/TxR and control cells. For the statistical analysis, SPSS for Windows, version 23.0 (SPSS, Chicago, IL, USA), was used. All experimental data were obtained by independently performing experiments at least three times and considered statistically significant when the *p*-value was lower than 0.05. Gene expression data of normal and cancer tissues were obtained from The Cancer Genome Atlas database (TCGA, https://cancergenome.nih.gov). The association between gene expression levels and the overall survival rate of breast cancer patients was evaluated using the Kaplan–Meier Plotter (http://kmplot.com/analysis).

## 5. Conclusions

CAP was shown to set the MCF-7/TxR cells back to the Tx-sensitive state, offering the potential application of CAP for the treatment of TxR cancer. At the molecular level, CAP recovered the expression of a set of genes that had been deregulated in the course of TxR. Among the genes, tumor related DAGLA and CEACAM1 were proven essential for the acquisition of resistance and for the recovery of sensitivity. These genes could be developed as diagnostic markers and could be molecular targets for the clinical treatment of TxR breast cancer.

## Figures and Tables

**Figure 1 cancers-11-02011-f001:**
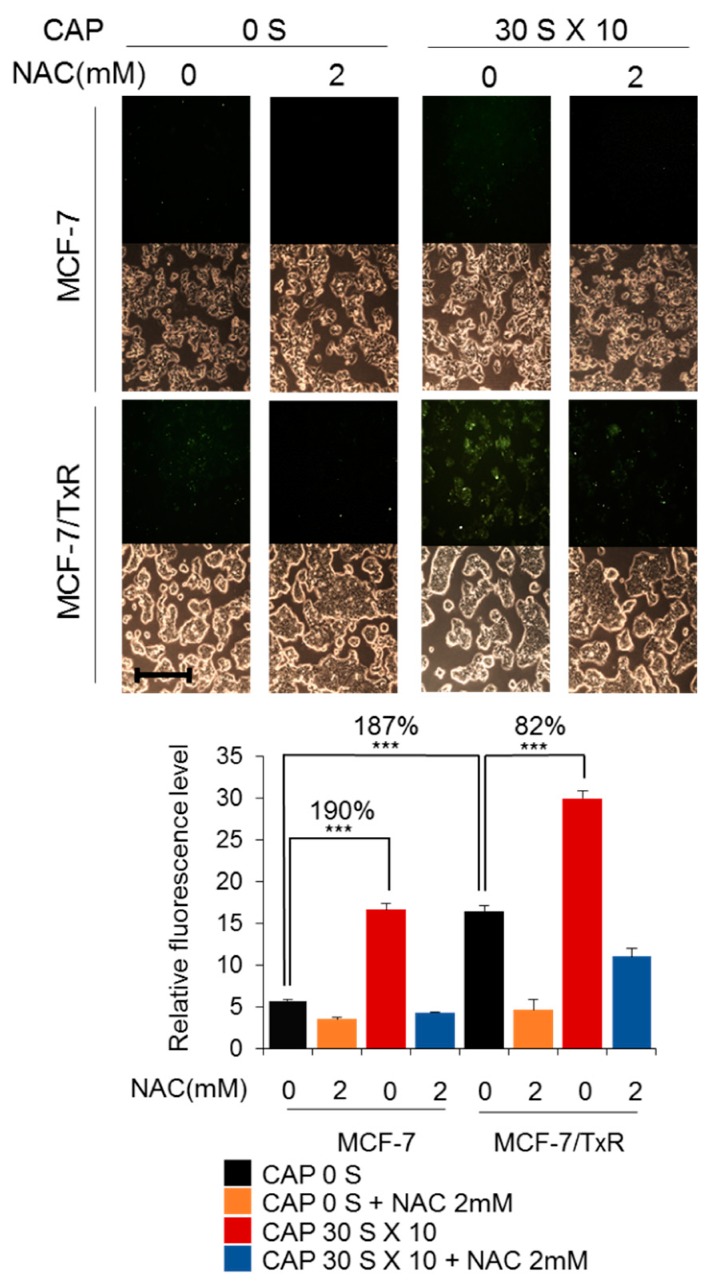
CAP increases ROS in MCF-7/TxR cells. Fluorescence microscope images were taken of MCF-7 and MCF-7/TxR cells by treating the cells with a fluorescent dye DCFH-DA after CAP treatment. N-acetylcysteine (NAC) was used to quench intracellular ROS. Monochrome images were obtained with a bright field microscope. Scale bar, 400 μm. *** *p* < 0.001.

**Figure 2 cancers-11-02011-f002:**
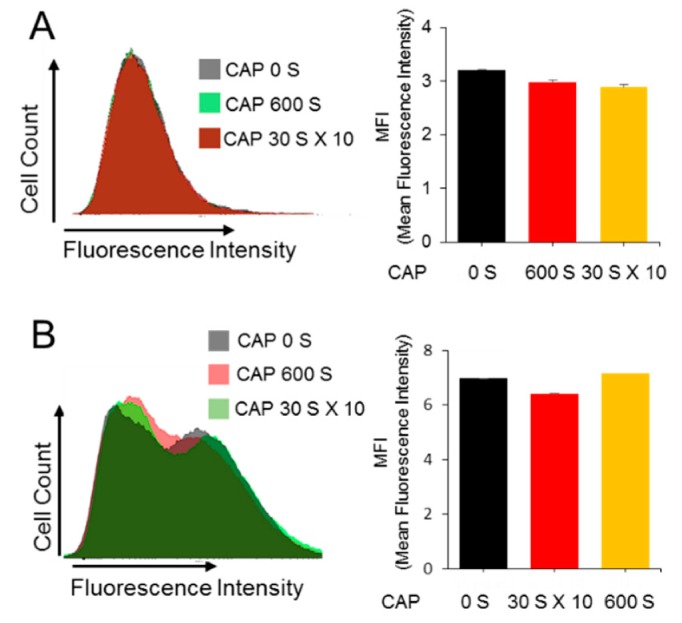
CAP does not affect uptake of Tx into MCF-7/TxR cells. MCF-7 and MCF-7/TxR cells were cultured in drug-containing media and treated with CAP. The uptake rate of doxorubicin (**A**) or Flutax-1 (**B**) in the MCF-7/TxR cells was examined by FACS, and the results are represented by bar graphs. All assays were performed in triplicate, and the results are expressed as mean ± SE.

**Figure 3 cancers-11-02011-f003:**
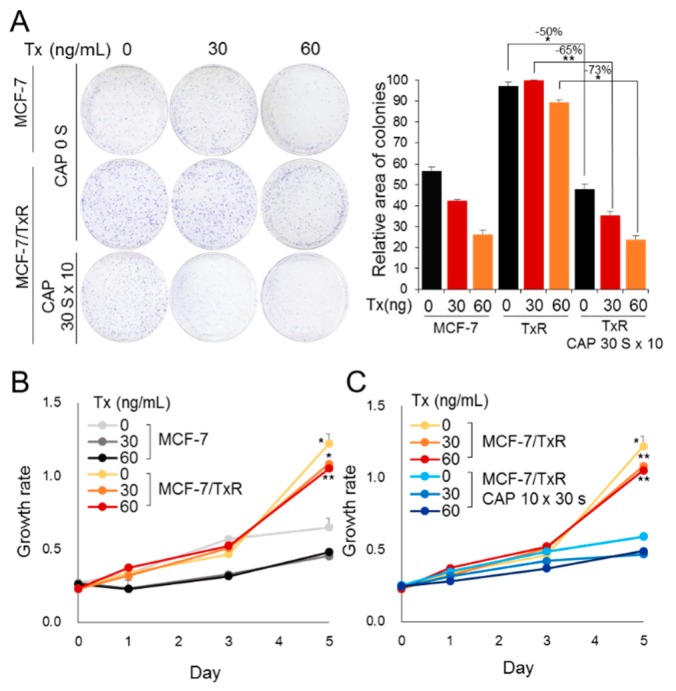
CAP sensitizes MCF-7/TxR cells to Tx. (**A**) The effect of CAP on the sensitivity of MCF-7 and MCF-7/TxR to Tx was examined by colony formation. The area of colonies is represented by a bar graph. (**B**) Effect of Tx on the growth rate of MCF-7/TxR vs. MCF-7. Cell growth was examined by CCK-8 assay. (**C**) Effect of CAP on growth rate of MCF-7/TxR in presence of Tx. All assays were performed in triplicate, and the results are expressed as mean ± SE. * *p* < 0.05, ** *p* < 0.01.

**Figure 4 cancers-11-02011-f004:**
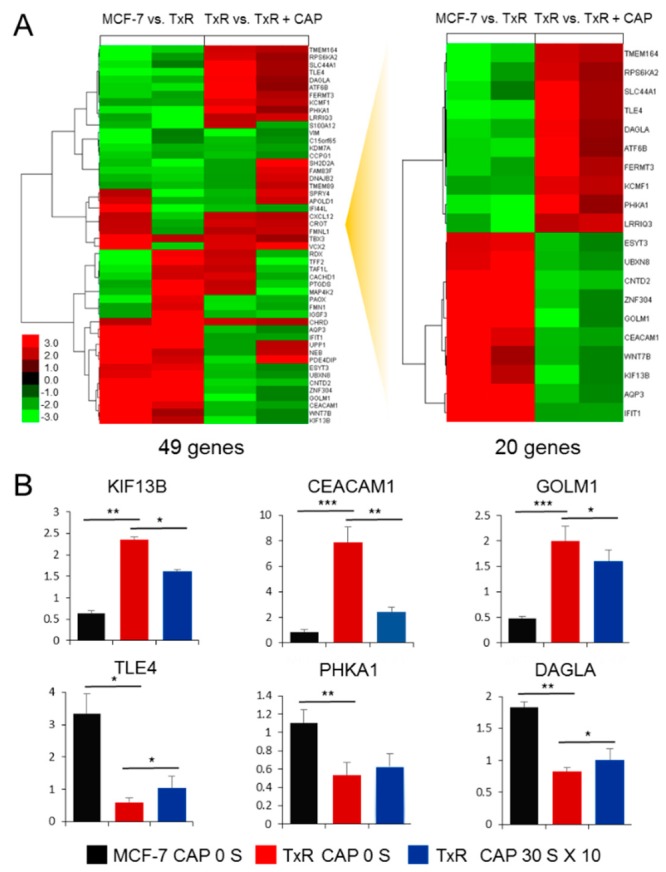
Clustering of genes affected by Tx and CAP in MCF-7 and MCF-7/TxR. (**A**) Heatmap analysis of 49 genes that exhibited expression changes (|fold change| ≥ 1.3) both in MCF-7 vs. MCF-7/TxR and MCF-7/TxR vs. CAP-treated MCF-7/TxR. Twenty genes showed opposite expression profiles at the two comparisons. Data are from expression arrays in duplicate. (**B**) qPCR of six genes that were selected from (**A**) showing upregulation in MCF-7 vs. MCF-7/TxR and downregulation in MCF-7/TxR vs. CAP-treated MCF-7/TxR (upper graphs), or vice versa (lower graphs). All assays were performed in triplicate, and the results are depicted as mean ± SE. * *p* < 0.05, ** *p* < 0.01, *** *p* < 0.001.

**Figure 5 cancers-11-02011-f005:**
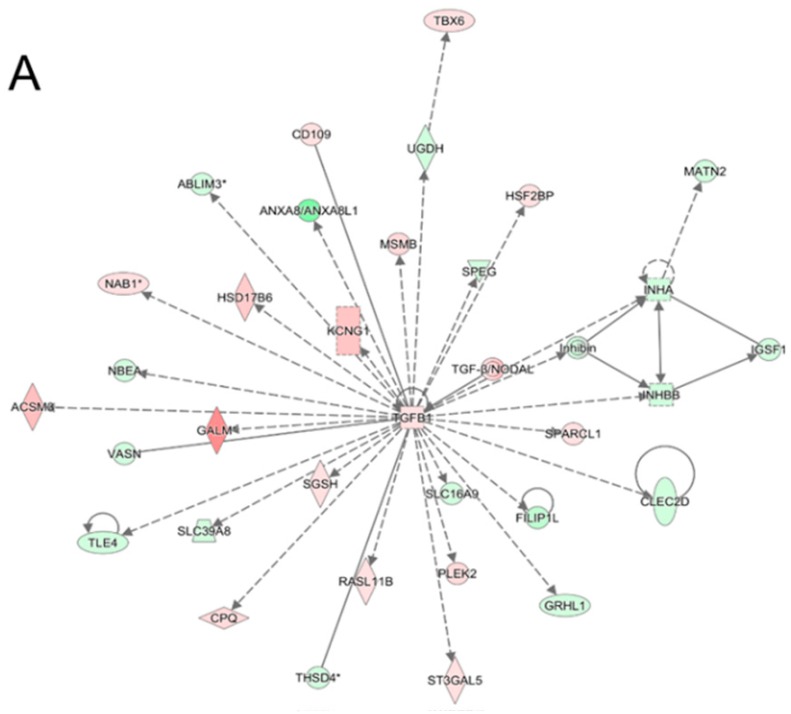
Highest confidence network of genes displaying altered expression levels by acquisition of TxR or by CAP. Most reliable IPA network of genes showing altered expression in MCF-7/TxR vs. MCF-7 (**A**) and CAP-treated MCF-7/TxR vs. MCF-7/TxR (**B**). The top networks are “Nutritional Disease, Organismal Injury and Abnormalities, and Carbohydrate Metabolism” (**A**) and “Embryonic Development, Nervous System Development and Function, and Organ Development” (**B**). Up- and downregulated genes are green and red, respectively, with the color intensity of the expression change. Dashed and solid lines denote direct and indirect interactions, respectively.

**Figure 6 cancers-11-02011-f006:**
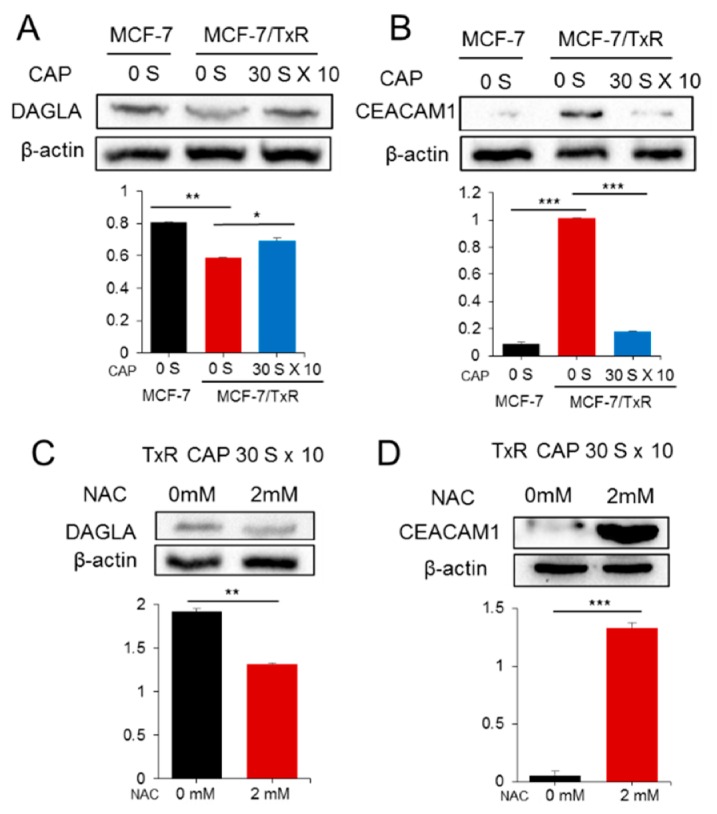
Protein expression of DAGLA and CEACMA1 in CAP-treated MCF-7/TxR cells. The protein expression of DAGLA (**A**) and CEACAM1 (**B**) in CAP-exposed MCF-7/TxR cells was examined by Western blot analysis. The protein expression of DAGLA and CEACAM1 showed a similar profile to that of microarray and qPCR. (**C**,**D**) The effect of CAP on the expression of DAGLA and CEACAM1 was examined after NAC treatment. NAC deteriorated the effect of CAP on the two proteins. The protein band in the Western blot was measured by a gel document system, and the levels are represented by bar graphs. All assays were performed in triplicate, and the results are depicted as mean ± SE. * *p* < 0.05, ** *p* < 0.01, *** *p* < 0.001.

**Figure 7 cancers-11-02011-f007:**
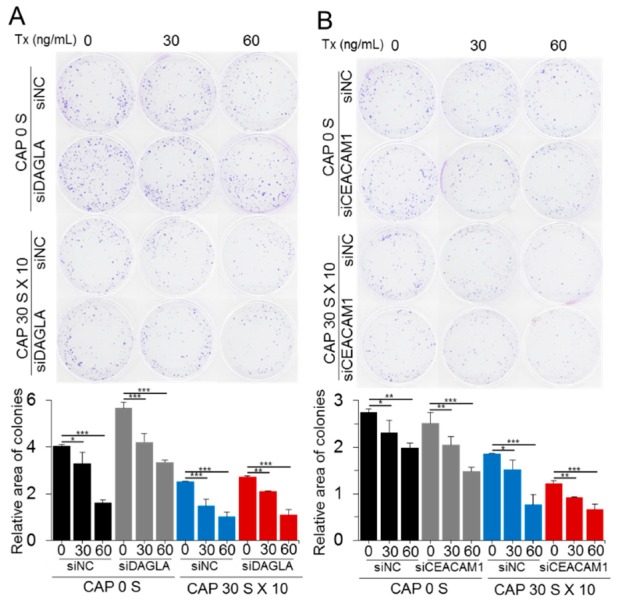
DAGLA and CEACAM1 are involved in the CAP-mediated recovery of sensitivity to Tx. Effect of DAGLA (**A**) and CEACAM1 (**B**) on the recovery of Tx sensitivity was determined by downregulating the genes using siRNA in MCF-7/TxR cells. The sensitivity to Tx was examined by colony formation assay after treating cells with CAP and Tx. siNC, negative-control siRNA. All assays were performed in triplicate, and the results are depicted as mean ± SE. * *p* < 0.05, ** *p* < 0.01, *** *p* < 0.001.

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
