# Peer review of "Cold Atmospheric Plasma Restores Paclitaxel Sensitivity to Paclitaxel-Resistant Breast Cancer Cells by Reversing Expression of Resistance-Related Genes"

_cancers, 2019, doi:10.3390/cancers11122011_

Round 1

Reviewer 1 Report

This paper by Park and co-workers presents the results of an in vitro study concerning the ability a DBD type of plasma device to restore the sensitivity of paclitaxel-resistant MCF-7 cells to paclitaxel (Tx) itself. The authors selected MCF-7 cells resistant to Tx (named MCF-7/TxR) and CAP treatment prior to Tx treatment restored sensitivity of MCF-7/TxR to Tx. Furthermore, the authors showed that gene expression is different between parental MCF-7 and MCF-7/TxR, but also between MCF-7/TxR and CAP-treated MCF-7/TxR. They identified 20 genes whose expression level in MCF-7/TxR is reversed by CAP treatment. Although the experiments were well conducted I don’t think that the authors provide experimental evidences that support their conclusion that is to say CAP restores sensitivity by reversing expression of resistance related genes. The link between sensitivity and reversion has not been clearly established. Their conclusion is based on the fact that the expression levels of 2 selected proteins (assessed by Western blotting), namely DAGLA and CEACAM1, were changed in MCF-7 compare to MCF-7/TxR (downregulated and upregulated, respectively) and that CAP treatment reverses these change to levels similar to those observed in parental MCF-7. But in any case, the authors demonstrated that these 2 proteins are indeed directly involved in the resistance/sensitivity to Tx. If down regulation of DAGLA expression (and inversely upregulation of CEACAM1) contributes to Tx resistance in MCF-7/TxR, what is the effect of down regulating DAGLA (and inversely up regulating CEACAM1) in MCF-7 and therefore sensitivity/resistance of these new engineered cell lines to Tx? The fact that expression level of these 2 genes changes upon CAP +/- NAC treatment suggests that their expression is likely regulated by ROS. This hypothesis has not been investigated, although I understand this is not the purpose of this paper, but most importantly not even discussed. In fact, in 2017, the same group published a similar paper (Lee et al. Free Radical Biology and Medicine 110 (2017) 280–290) in which they showed that their CAP device restores tamoxifen sensitivity in resistant MCF-7 breast cancer cell in a way that is very similar to what is shown in the present submitted manuscript. This prompts me to believe that the DBD CAP device used by the authors has this unique power/property of restoring the sensitivity of drug-resistant cells to the drug that was used to select resistant cells by reprogramming gene expression in these cells. I’m a little skeptical about this hypothesis. I think this should also be discussed. No doubt that long term drug treatment led to epigenetic changes/mutations that explain the transition from MCF-7 to MCF-7/TxR. And no doubt that CAP treatment itself also modifies gene expression. Again, my main concern is the lack of clear evidence between restoration of sensitivity and change in the expression of some specific genes in response to CAP. Insofar as CAP treatment can be seen as a mixture of ROS and RNS, what is the impact of reconstitution buffer (composed of H2O2, NO2 and NO3 at concentrations generated by the DBD CAP device in the experimental conditions described in the manuscript) on the sensitivity of MCF-7/TxR cells to Tx and on gene expression? Indeed, another explanation could be that CAP-induced intracellular and extracellular production of ROS increases the toxicity of the drugs without changing its uptake but independently of change of gene expression.

Other comments:

-lines 32-33: typing error « A high ratio of cancer patients who receive chemotherapy present with drug resistance”

- lines 73-76: What is the interest of the ROS experiments as it is not really used/discussed in the rest of the manuscript? Why looking at 24h post CAP and not immediately? What is the sensitivity of the 2 cell lines to Tx??

- Lines 76-77: “To determine whether the recovery of sensitivity by CAP comes from the increased pumping…” But the recovery experiments are only shown in Figure 3. Why showing this set of experiments (Figure 2) before the sensitivity??

- Lines 83-86 Figure 1: Why the experiments were not performed by flow cytometry which is a more sensitive and faster assay? Especially, the basal level of ROS can be clearly measured and quantified by flow cytometry compare to fluorescence microscopy.

- line 92: When Tx was added to the culture? Immediately or later after the last CAP treatment??

- line 95: Why "Surprisingly"?? The authors have already shown this effect using Tamoxifen (Lee et al., FRBM 2017) so is that really surprising?

- Lines 104-108 Figure 3 and line 226: What is measured? The area of colony or the number of colonies? Furthermore, what is the volume of medium treated in 60mm culture dish? Similarly, what is the volume of medium treated per well in 96 well plate?? I guess the difference is probably quite important (2-3 ml for 60 mm dish versus 200-300 µl for 96 well plates). Therefore, the concentration of CAP-induced ROS/RNS is more important in 96 well plate compare to 60 mm dish. It is then difficult to compare Figure 3A (60 mm dish) and Figures 3B-3C (96 well plate). Why untreated MCF-7 cells stop to grow after 3 days of culture in Figure 3B??

- Line 143: “CEACAM1 and DAGLA mediated the restoration of sensitivity to Tx” To my knowledge, the authors did not demonstrate that CEACAM1 and DAGLA mediate the restoration of sensitivity to Tx.

-Lines 155-162 Figure 6: What is the effect of NAC on CEACAM and DAGLA in absence of CAP?

- Lines 173-174: “In fact, the MCF-7/TxR cells showed a higher growth rate than MCF-7 and a higher level of death induction by CAP.” The higher level of growth rate is based on Figure 3B in which MCF-7 cells stopped to grow after day 3?? Although, there is no data on the level of cell death induced by CAP treatment in parental MCF-7??

Author Response

The authors deeply appreciate the reviewer’s extensive comments which are believed to improve the quality of the manuscript. For convenience, the authors summarized the comments into four categories and responded as below.

(Reviewer’s comments) What is the effect of deregulating DAGLA and CEACAM1 in MCF-7 and therefore sensitivity/resistance of these new engineered cell lines to Tx?

(Authors’ response) To concrete the authors’ claim that DAGLA and CEACAM1 are responsible for the recovery of sensitivity to Tx, downregulation of the two genes was induced using siRNAs and the effect on sensitivity was monitored via colony formation assay. As results, transfection of siRNA for DAGLA deteriorated the recovery effect of plasma while siRNA for CEACAM1 ameliorated the effect, confirming the contribution of the genes to sensitivity recovery to Tx. The data is added in the Figure. 7.

(Reviewer’s comments) The fact that expression level of these 2 genes changes upon CAP +/- NAC treatment suggests that their expression is likely regulated by ROS. Although I understand this is not the purpose of this paper, this hypothesis has to be discussed.

(Authors’ response) NAC is an ROS inhibitor and thereby widely used for the chelation of ROS inside cells. In this study, the level of ROS was decreased by NAC treatment in the MCF-7 as well as its Tx-resistant cells. The effect of plasma on the expression of DAGLA and CEACAM1 was diminished by NAC treatment. But, cautious explanation is needed because other reactive species such as NOS in addition to ROS could be also involved in the regulation. This was discussed in the revised manuscript.

(Reviewer’s comments) DBD CAP device used by the authors has this unique power/property of restoring the sensitivity of drug-resistant cells to the drug that was used to select resistant cells by reprogramming gene expression in these cells. I think this should also be discussed.

(Authors’ response) This study was performed whether the application of CAP could be expanded to a wider range of drug-resistant cancer cells. CAP successfully recovered sensitivity to Tx as like as Tam. To generalize the concept that CAP recovers sensitivity to a wide spectrum of drugs further evaluation is needed in those resistant cells. Meanwhile, judging from the fact that different gene sets are regulated in the two resistant cells by CAP, the molecular mechanism is not common and therefore how CAP induces sensitivity in specific drug-resistance cells should be carefully evaluated. This speculation was discussed in the revised manuscript.

(Reviewer’s comments) What is the impact of reconstitution buffer (composed of H2O2, NO2 and NO3 at concentrations generated by the DBD CAP device in the experimental conditions described in the manuscript) on the sensitivity of MCF-7/TxR cells to Tx and on gene expression?

(Authors’ response) As suggested by the reviewer, it would be helpful if experiments is performed using a reconstituted buffer. This approach would be supplemental with the one using the ROS or RNS inhibitors. This consideration was added in the section of inhibitor while revising the discussion of the manuscript.  

Other comments:

-lines 32-33: typing error « A high ratio of cancer patients who receive chemotherapy present with drug resistance”

(Authors’ response) The words ‘present with’ was corrected to ‘acquire’.

- lines 73-76: What is the interest of the ROS experiments as it is not really used/discussed in the rest of the manuscript? Why looking at 24h post CAP and not immediately? What is the sensitivity of the 2 cell lines to Tx??

(Authors’ response) As indicated in the Authors’ response #2 and #4 above, discussion about the involvement of ROS was added. In all experiments to analyze RNA and protein expression, cells were incubated for 24 h after CAP treatment to give enough time for gene expression change. ROS level was monitored 24 h after CAP treatment, which was the same time gap as the RNA and protein analysis. Sensitivity of MCF-7 and its Tx-resistant cells is indicated in Figure 3.

- Lines 76-77: “To determine whether the recovery of sensitivity by CAP comes from the increased pumping…” But the recovery experiments are only shown in Figure 3. Why showing this set of experiments (Figure 2) before the sensitivity??

(Authors’ response) The recovery of sensitivity may come from either alteration of drug transport across cell membrane or regulation of gene expression. At first, Tx levels were measured to examine the possibility of transport rate change by CAP before elucidating the molecular mechanism. The miss-typed word, ‘increased’ was corrected to ‘decreased’.

- Lines 83-86 Figure 1: Why the experiments were not performed by flow cytometry which is a more sensitive and faster assay? Especially, the basal level of ROS can be clearly measured and quantified by flow cytometry compare to fluorescence microscopy.

(Authors’ response) The authors wanted see the fluorescence visibly under a microscope. As indicated by the reviewer, the method is less sensitive but used as it did not hamper to detect the changed ROS level.

- line 92: When Tx was added to the culture? Immediately or later after the last CAP treatment??

(Authors’ response) Tx was added 24 h after CAP treatment. This was edited in the Materials and Methods.

- line 95: Why "Surprisingly"?? The authors have already shown this effect using Tamoxifen (Lee et al., FRBM 2017) so is that really surprising?

(Authors’ response) It is notable that CAP resets the resistant cells’ sensitivity to Tx as like as Tam even though the acting mechanism of the two drugs are different. ‘Surprisingly’ was replaces with ‘Notably’.

- Lines 104-108 Figure 3 and line 226: What is measured? The area of colony or the number of colonies? Furthermore, what is the volume of medium treated in 60mm culture dish? Similarly, what is the volume of medium treated per well in 96 well plate?? I guess the difference is probably quite important (2-3 ml for 60 mm dish versus 200-300 µl for 96 well plates). Therefore, the concentration of CAP-induced ROS/RNS is more important in 96 well plate compare to 60 mm dish. It is then difficult to compare Figure 3A (60 mm dish) and Figures 3B-3C (96 well plate). Why untreated MCF-7 cells stop to grow after 3 days of culture in Figure 3B??

(Authors’ response) The area of colonies was measured using ImageJ software program. 2 ml and 100 μl of medium was added per 60 mm dish and a well of 96 well plate. Analyzing colony and medium volume were specified in the Materials and Methods. The authors agree with the reviewer’s opinion that 96 well is more efficient than 60 mm dish in terms of ROS/RNS level, so 96 well plate is recommended to improve the efficacy of plasma. But, figure 3 is designed to examine efficacy of Tx to recover sensitivity in the two types of cell culture.

- Line 143: “CEACAM1 and DAGLA mediated the restoration of sensitivity to Tx” To my knowledge, the authors did not demonstrate that CEACAM1 and DAGLA mediate the restoration of sensitivity to Tx.

(Authors’ response) This is the major issue raised by the reviewer and the authors performed additional experiments. The authors’ response is described in the Authors’ response #1 to the major comments above.

-Lines 155-162 Figure 6: What is the effect of NAC on CEACAM and DAGLA in absence of CAP?

(Authors’ response) It could be anticipated that NAC alone induces a decrease of DAGLA as it suppressed the increasing effect of CAP. In the case of CEACAM1, the situation is vice versa. Nonetheless, to refine the experiment treatment of NAC alone is recommended to refine the experiment.

- Lines 173-174: “In fact, the MCF-7/TxR cells showed a higher growth rate than MCF-7 and a higher level of death induction by CAP.” The higher level of growth rate is based on Figure 3B in which MCF-7 cells stopped to grow after day 3? Although, there is no data on the level of cell death induced by CAP treatment in parental MCF-7

(Authors’ response) The higher growth rate of MCF-7/TxR and death induction can be found at the colony formation assay. The data for cell death of MCF-7 by CAP was added in the Supplementary Figure S1.

Reviewer 2 Report

In their manuscript „Cold atmospheric plasma restores paclitaxel sensitivity to paclitaxel-resistant breast cancer cells by reversing expression of resistance-related genes” Park and co-workers show that cold atmospheric plasma (CAP) could recover paclixatel (Tx) sensitivity in a Tx-resistance breast cancer cell line.

These are very interesting result that are worth publishing in Cancers. There are, however, few questions that should be addressed in revision.

The main problem is, that in all experiments presented here CAP treatment results are only presented in Tx resistant MCF7 cell. The experiments must have CAP effects in “normal” MCF7 as controls. How does CAP affect the cellular response? By ROS production? Most likely, but the also a control must be included that should diminish the CAP effects (colony formation, gene regulation) by NAC treatment. In Fig. 1 it is not clear how the results were quantified. What does “relative fluorescence level” mean? Relative to what parameter? In Fig. 3/4 MCF7 CAP control is missing (see comment 1.) The authors claim that “CEACAM1 and DAGLA mediated the restoration of sensitivity to Tx”. They should present recombinant data (knockout CEACAM1 or DAGLA) to present evidence for that statement. Otherwise the authors could mention that CEACAM1 and DAGLA are involved in the process.

Author Response

(Authors’ response)

- Experiments of CAP effect on MCF-7 control cell was performed and added in the Supplementary Figure S1.

- The fluorescence level was quantified using a fluorescence plate reader and the experimental protocol was detailed in the Materials and Methods.

- In the revised manuscript, downregulation of DAGLA and CEACAM1 were induced using siRNAs and the effect on sensitivity was monitored by colony forming assay. As results, transfection of siRNA for DAGLA deteriorated the recovery effect of plasma while siRNA for CEACAM1 ameliorated the effect, confirming the contribution of the genes to sensitivity recovery to Tx. The data is added in the Figure 7.

Reviewer 3 Report

The authors show that CAP can increase ROS in MCF-7/TxR cells. Also, the expression of their drug resistance-related genes, DAGLA and CEACMA1, can be regulated by ROS inhibitor, NAC. The authors should apply NAC in Figure 3 to prove the relation between ROS level and CAP-induced effects. The authors show a bit of a laundry list of commonly characterized pathways. How do these pathways relate to drug resistance or more specific paclitaxel resistance and/or ROS regulation? Maybe these are the relevant pathways, but not clear to me.   The authors claim that CEACAM1 and DAGLA can mediate the restoration of sensitivity to paclitaxel in MCF-7/TxR cells. CEACAM1 does have marked changes in expression between different treatments; however, the differential level of DAGLA expression is not convincing. In addition, the authors have to conduct knock-in and knock-out experiments on both genes to make this conclusion. I agree that one of the limitations of this study is the lack of in vivo results. However, on top of that, the authors must improve their results to prove that CEACAM1 and DAGLA are really the critical genes that involve in the CAP-restored paclitaxel sensitivity.

Author Response

(Authors’ response) The authors deeply appreciate the reviewer’s extensive comments which are believed to improve the quality of the manuscript. For convenience, the authors summarized the comments into three categories and responded as below.

The authors should apply NAC in Figure 3 to prove the relation between ROS level and CAP-induced effects.

(Authors’ response) Because CAP is composed of diverse reactive species other than ROS, such as RNS, electrons, and UV, variable inhibitors should be used to comprehend the relation between the overall CAP composites and CAP effect. Therefore Figure 3 is designed to focus on examining the efficacy of CAP to recover sensitivity to Tx in the Tx-resistant MCF-7.

How do these pathways relate to drug resistance or more specific paclitaxel resistance and/or ROS regulation?

(Authors’ response) The pathway is the highest confidence network constructed using the Ingenuity pathway analysis system from a set of genes that were significantly deregulated in the course of resistance acquisition and CAP treatment. DAGLA and CEACAM1 were revealed to contribute to the drug sensitivity. Other genes in the pathway are also considered to have potential to regulate the drug resistance and sensitivity. This fact is further discussed in the revised manuscript.

The authors must improve their results to prove that CEACAM1 and DAGLA are really the critical genes that involve in the CAP-restored paclitaxel sensitivity.

(Authors’ response) To concrete the authors’ claim that DAGLA and CEACAM1 are responsible for the recovery of sensitivity to Tx, downregulation of the two genes were induced using siRNAs and the effect on sensitivity was monitored via colony formation assay. As results, transfection of siRNA for DAGLA deteriorated the recovery effect of plasma while siRNA for CEACAM1 ameliorated the effect, confirming the contribution of the genes to sensitivity recovery to Tx. The data is added in the Figure 7.

Round 2

Reviewer 1 Report

I still have a major problem with the data provided by the authors and their conclusion. Although I fully agree that CAP restores sensitivity to Tx, I still believe that their experiments do not support their conclusions, especially the link between restoration of sensitivity and reversing expression of resistance-related genes.

1) My first criticism concerns Figure 3 especially the colony forming assay. In the M&M it is clearly written “For the colony formation assay, 3×103 cells of MCF-7 and MCF-7/TxR were seeded in 60-mm culture dishes with 2 ml medium and treated with CAP followed by exposure to either Tx or DMSO as a vehicle control 24 h later”. A colony forming assay (also called clonogenic assay) is an assay that monitor the ability of a single cell to divide and to form a colony of at least 50 cells [Franken et al. Clonogenic assay of cells in vitro. Nat Protoc. 2006;1(5):2315-9]. According to the M&M, 3000 cells were seeded per 60-mm dish. The first parameter that needs to be indicated is the plating efficiency for each cell line. Indeed, the title of the y-axis in Figure 3A is “number of colonies”. In absence of any treatments (No CAP, no Tx), these values are ~55 for MCF7 and 95 for TxR. As 3000 cells were seeded per dish, this means that the plating efficiency for MCF7 is (55/3000)x100 = 1.83% and for TxR (95/3000)x100 = 3.16%. These values are extremely low. But a careful look at the plates displayed in Figure 3A strongly suggests there are more than 55 or 95 colonies per dish in untreated cells. So my question is: what is exactly shown in Figure 3A??? Again, according to the authors, the colony area was measured using the ImageJ software program and what is indicated is the number of colonies?? What was the purpose of measuring the colonies area??? This value will depend on the cell growth, division time, and as shown in Figure 3B the 2 cell lines showed different cell division time. Usually, if a cell line growths slower than another one, the best is to wait few more days for the slow dividing cells before fixing and staining the cells so that the size of the colonies are similar between cell lines.

            In Figure S1-A, what is the title of the y-axis? In Figure S1-B what means “early” and “late”?? At which time points the data were collected? Why apoptosis is induced in untreated cells (from few % at early time point to 21% at late time point)?

2) My second criticism concerns Figure 7. Again, what is the purpose of measuring the area of colonies??? Two cell lines could have the same response to a stress in a term of colonies formation but one can give rise to smaller colonies than the other. And the conclusion will not be that one is more sensitive/resistant than the other. Furthermore, Figure 7 shows that down-regulation of DAGLA in MCF7/TxR leads to a higher rate of proliferation in those cells, as indicated by the authors (lanes 155-156 of the revised manuscript). But, according to Figure 3, MCF7/TxR cells are already proliferating faster than Tx sensitive parental MCF7. Collectively these results suggest that proliferation rate of MCF7/TxR siDAGLA > proliferation rate of MCF7/TxR > proliferation rate of MCF7 implicating DAGLA in controlling the rate of proliferation in MCF7 (low expression high proliferation rate, high expression low proliferation rate). Moreover, the “size of the colonies” in MCF7/TxR siNC and in MCF7/TxR siDAGLA is the same after CAP treatment +/- Tx treatment. One interpretation is that CAP treatment favors Tx toxicity irrespective of DAGLA status. The same conclusion can be driven with CEACAM1.

Figure S2. The authors show the mRNA level of DAGLA and CEACAM1 in MCF7/TxR siNC, MCF7/TxR siDAGLA and in MCF7/TxR siDAGLA. What is the protein level of DAGLA and CEACAM1 in those cells?? What is the level at 0nM Tx??

Reviewer 3 Report

The authors claimed that CAP recovered paclitaxel sensitivity of paclitaxel-resistant breast cancer cells by resetting the expression of multiple drug resistance-related genes, with an emphasis on DAGLA and CEACAM1. (1) CAP restored the drug sensitivity by upregulating DAGLA and downregulating CEACAM1. Therefore, it makes sense to know down DAGLA by siRNA. However, in Fig. 7, the DAGLA knockdown only increased colony formation without CAP. The drug sensitivity looks no markedly change between siControl and SiDAGLA cells under CAP condition. (2) As aforementioned, CAP restored the drug sensitivity by downregulating CEACAM1. It's hard to understand that the authors perform a CEACAM1 knockdown instead of CEACAM1 overexpression to prove its role in CAP-induced drug sensitivity. In addition, n Fig. 7, the CEACAM1 knockdown slightly increased the drug sensitivity of MCF-7/TxR cells both with and without CAP. Taken together, the results in Fig 7 is hard to prove that CAP-induced drug sensitivity is mediated by DAGLA and CEACAM1.  

Round 3

Reviewer 1 Report

I appreciate that the authors change the initial sub-heading "CEACAM1 and DAGLA mediated the restoration of sensitivity to Tx" to "CEACAM1 and DAGLA are regulated during the restoration of sensitivity to Tx by CAP".

Furthermroe, I suggest to the authors to add the following reference that describes the software used to quantify colony formation in clonogenic assays. It could be useful to readers.

Ref: Guzman C, Bagga M, Kaur A, Westermarck J, Abankwa D (2014) ColonyArea: An ImageJ Plugin to Automatically Quantify Colony Formation in Clonogenic Assays. PLoS ONE 9(3): e92444. doi:10.1371/journal.pone.0092444